# Detecting Faking on Self-Report Measures Using the Balanced Inventory of Desirable Responding

**Walter P. Vispoel** [1,*] , **Murat Kilinc** [1] and **Wei S. Schneider** [2]

1. Department of Psychological and Quantitative Foundations, University of Iowa, Iowa City, IA 52242, USA; murat-kilinc@uiowa.edu
2. Cambium Assessment, Washington, DC 20007, USA; wei.schneider@cambiumassessment.com
* Correspondence: walter-vispoel@uiowa.edu

**Abstract:** We compared three methods for scoring the Balanced Inventory of Desirable Responding (BIDR) to detect faked responses on self-report measures: (1) polytomous, (2) dichotomous emphasizing exaggerating endorsement of socially desirable behaviors, and (3) dichotomous emphasizing exaggerating denial of such behaviors. The results revealed that respondents on average were able to fake good or fake bad and that faking markedly affected score distributions, subscale score intercorrelations, and overall model fits. When using the Impression Management scale, polytomous and dichotomous exaggerated endorsement scoring were best for detecting faking good, whereas polytomous and dichotomous exaggerated denial scoring were best for detecting faking bad. When using the Self-Deceptive Enhancement scale, polytomous and dichotomous exaggerated endorsement scoring again were best for detecting faking good, but dichotomous exaggerated denial scoring was best for detecting faking bad. Percentages of correct classification of honest and faked responses for the most effective methods for any given scale ranged from 85% to 93%, with accuracy on average in detecting faking bad greater than in detecting faking good and greater when using the Impression Management than using the Self-Deceptive Enhancement scale for both types of faking. Overall, these results best support polytomous scoring of the BIDR Impression Management scale as the single most practical and efficient means to detect faking. Cut scores that maximized classification accuracy for all scales and scoring methods are provided for future use in screening for possible faking within situations in which relevant local data are unavailable.

**Keywords:** socially desirable responding; fake detection; self-reports; scoring methods; personality assessment; Balanced Inventory of Desirable Responding; classification analysis; reliability; validity

## 1. Introduction

Socially desirable responding has long posed a threat to the valid interpretation of results from self-report questionnaires. A common way to detect such responding is to include items to measure it directly within the target questionnaire(s) of interest or within a companion measure to those questionnaires. In the study reported here, we evaluated the accuracy of scores from Version 6 of the Balanced Inventory of Desirable Responding (BIDR; [1–3]) in detecting instances of faking good and faking bad using a variety of scoring methods.

## 2. Background

### 2.1. Socially Desirable Responding

The beginning of objectively scored non-cognitive assessments within the psychometric research literature is often traced back to the creation of Woodworth's Personal Data Sheet in 1919 (WPDS; aka Woodworth's Psychoneurotic Inventory; see, e.g., [4]). Following that landmark development, construction and uses of such measures have proliferated. New objectively scored measures of personality, self-concept, interests, and

attitudes emerge with each passing year, moving increasingly from traditional paper-and-pencil to computerized administration formats. Such instruments are used routinely for diagnosis, placement, theory building, validation, prediction, and selection purposes.

However, one of the most serious drawbacks to such non-cognitive measures is that they are susceptible to response biases that can undermine valid interpretation of results. Pervasive among such biases is socially desirable responding. Socially desirable responding reflects unconscious or willful tendencies to strongly endorse socially acceptable behaviors (e.g., fake good), or to deny and underreport the same behaviors (e.g., fake bad) if something is gained by doing so. A common way to address such problems is to administer measures intended to measure socially desirable responding directly along with the target questionnaire(s) of interest. Early examples of such measures include the K and L scales from the Minnesota Multiphasic Personality Inventory [5,6], Marlowe-Crowne Social Desirability Scale [7], Edwards Social Desirability Scale [8], Eysenck Lie Scale [9], Martin-Larsen Approval-Motivation Scale [10], Jacobson-Kellogg Social Desirability Inventory [11], and Self- and Other-Deception Questionnaires [12].

Prominent instruments, such as the Edwards Social Desirability Scale and the Marlowe-Crowne Social Desirability Scale, were based on a unidimensional view of socially desirable responding. However, subsequent factor analytic studies of responses to scales measuring social desirability embedded within established instruments and stand-alone measures provided convincing evidence that socially desirable responding consisted of at least two discernable components consisting of self-deception and conscious faking components [1,13–21]. Paulhus [1–3] subsequently labeled these components as Self-Deceptive Enhancement (SDE) and Impression Management (IM). SDE represents an unconscious bias of over-reporting oneself in a favorable light, whereas IM represents more intentional attempts to distort responses to win other people's approval or disapproval when such responses can lead to personal gain. Accordingly, IM is generally considered a more serious threat to the validity of results.

## 2.2. The Balanced Inventory of Desirable Responding

Paulhus (1–3, 18–20) developed the Balanced Inventory of Desirable Responding (BIDR) as a stand-alone inventory to measure both SDE and IM. The BIDR is used primarily for four purposes. Three are catered to norm-referencing of scores (measure validation, outcome assessment, and statistical control), and one is catered to criterion referencing (flagging instances of invalid responding; see Paulhus [1]). Versions of the BIDR are available using either 5- or 7-point response metrics [1–3] with the lowest scale point for both metrics labeled "not true", and the highest labeled "very true".

**Scoring the BIDR.** Paulhus [1] describes two basic ways for scoring the BIDR: (a) polytomously, based on the original item scores, and (b) dichotomously, to reflect only exaggerated endorsement of socially desirable behaviors. With dichotomous scoring, 1 is assigned to exaggerated high endorsement of a socially desirable behavior within an item, and 0 is assigned to low to moderate endorsement. Between these two methods, Paulhus strongly advocates dichotomous scoring. As a result, this is the most common way for scoring the BIDR in research studies and practice.

However, a problem with Paulhus's recommended dichotomous scoring procedure is that it is ill suited for flagging faked-bad responses because only exaggerated endorsement of socially desirable behaviors is considered. For example, a respondent who is neutral or moderately endorses socially desirable behaviors could receive scores of 0 to all items, and therefore be flagged for faking bad. Vispoel and Kim [22], also see ref. [23] addressed this problem by developing an alternative dichotomous scoring procedure in which only exaggerated denial of socially desirable behaviors was scored as 0 and other responses as 1. They found that their alternative dichotomous method provided better discrimination at low construct levels and more reliable cut scores for flagging faked-bad responses than did original dichotomous scoring. Across these studies [22,23], findings were replicated

for dichotomous scores derived from either 5-point or 7-point polytomous scores, but no faking conditions were examined.

Research studies in which dichotomous and polytomous scoring have been compared under conditions of honest responding have almost uniformly shown that polytomous scoring provides superior evidence of psychometric quality for norm-referencing purposes [22–29]. For example, across these studies, reliability indices (alpha, split-half, test-retest, generalizability coefficients) and convergent validity coefficients were consistently higher for polytomous than for dichotomous scoring. Paulhus likely advocated dichotomous scoring of the BIDR as an intuitively appealing way to spot instances of faking good because only exaggerated endorsement of socially desirable behaviors contributes to high scores. However, dichotomized scoring inherently results in loss of information that lowers discrimination among responses by reducing the number of possible scale points for subscale scores. It also produces skewed distributions with predominantly low scores because most respondents do not provide extreme responses (see, e.g., [30]). With the 7-point scale metric and 20 items per subscale, dichotomous scoring reduces a 20 to 140 range of 121 possible points to a 0 to 20 range with 21 possible points. The severe restriction of the range from dichotomous scoring of the BIDR has undoubtedly contributed to lower reliability and convergent validity indices in comparison with polytomous scoring observed in the studies cited here [22–30].

**Detecting faking using the BIDR.** Even though polytomous scoring of the BIDR has yielded better psychometric indices of reliability and validity than has dichotomous scoring, the same might not be true for detecting faking. Unfortunately, very limited research has been conducted comparing BIDR scoring methods in fake detection, and what has been done is limited to one or two scoring methods and has excluded dichotomous exaggerated denial scoring altogether. In perhaps the earliest and most influential of these studies, Paulhus et al. [31] administered the BIDR to 370 undergraduates from a large research university and derived results using traditional dichotomous scoring. Participants were assigned at random to seven conditions: (1) *control*: "respond honestly", $n = 132$; (2) *fake best*: "fake the best candidate", $n = 48$; (3) *fake good*: "fake good without arousing suspicion", $n = 44$; (4) *play up*: "play up your good points", $n = 49$; (5) *fake modest*: "be somewhat modest in your answers", $n = 17$; (6) *fake bad*: "fake bad without arousing suspicion, $n = 37$; and (7) *fake worst*: "fake the worst possible candidate", $n = 43$. Statistically significant mean differences among groups were detected when using the IM scale but not the SDE scale. In comparison to the control (answer honestly) group ($M = 5.3$), IM means were significantly higher in the fake best ($M = 14.1$), fake good, ($M = 12.0$), and play up ($M = 10.8$) groups; significantly lower in the fake bad ($M = 2.5$) and fake worst group ($M = 1.1$); and not reliably different in the fake modest group ($M = 5.1$).

Most subsequent studies of faking using the BIDR focused on either traditional dichotomous [32–34] or polytomous scoring [35,36] alone and yielded results in line with Paulhus et al. [31] in which scores were typically higher under fake-good conditions and lower under fake-bad conditions in comparison to normal or honest responding. As part of a study cited earlier in which scoring methods were systematically compared, Stöber et al. [27] administered a German language translation of the BIDR [37,38] to 55 high school students randomly assigned to two conditions: fake-good ("make as good an impression as possible"; $n = 28$) and fake-bad ("make as bad an impression as possible"; $n = 27$). Each participant completed the BIDR honestly first, and then again in the targeted faking direction. Under fake-good instructions, IM and SDE scores displayed statistically significant increases for both polytomous and traditional dichotomous scores. Under fake-bad instructions, results revealed a statistically significant decrease for IM polytomous scores. However, contrary to expectations, traditional dichotomous scores yielded a small but non-significant decrease for IM but a significant increase for SDE, thereby highlighting flaws in traditional dichotomous scoring for detecting faking bad.

In the most recent study comparing scoring methods in detecting faking on the BIDR, Asgeirsdottir et al. [25] administered an Icelandic translation of the BIDR [39] at random

to two groups of university students: (1) "answer normally" (*n* = 258) and (2) "respond as socially desirable as possible" (*n* = 213). Means between groups were compared using polytomous and traditional dichotomous scoring. Consistent with most previous studies, statistically significant differences favoring the faking group were found for both IM and SDE using both scoring methods.

Overall, results from the studies led by Paulhus [31], Stöber [27], and Asgeirsdottir [35] have revealed that respondents on average are able to fake good and fake bad when instructed to do so, but more so for the IM than for the SDE scale. However, these studies are limited in several important ways. First, the samples for experimental groups were often modest in size (e.g., 17–49 in Paulhus et al. [31] and 27–28 in Stöber et al. [27]). Second, only traditional dichotomous endorsement scores were analyzed by Paulhus et al. [31] and all studies excluded alternative dichotomous scoring catered to detecting faking bad. Third, effect sizes for mean differences between the control and experimental groups were not reported. Fourth, results for Paulhus et al. [31] were based on the five-point rather than more commonly used 7-point original response metric. Fifth, cut scores were not reported for flagging faked responses. Sixth, classification accuracy indices were not reported for distinguishing honest and faked responses. Finally, measures were completed in paper rather than computer form.

## 3. Purpose

In the study reported here, we sought to overcome the limitations of previous research by including large and equal size experimental and control groups; comparing polytomous, traditional dichotomous (exaggerated endorsement), and alternative dichotomous (exaggerated denial) scoring of the BIDR in psychometric properties; computing effect size indices between honest and faking groups using the more common 7-point original response metric; deriving cut scores to flag instances of faking for possible future use; determining the accuracy of those cut scores in classifying honest and faked responses; administering all measures on computer; and identifying scales and scoring procedures that best detect each type of faking.

## 4. Methods

### 4.1. Measure, Sample, and Procedures

The BIDR includes two 20-item subscales that measure Impression Management (IM) and Self-Deceptive Enhancement (SDE). Items are stated as propositions and rated on a 7-point response metric (1 = not true, 4 = somewhat true, 7 = very true). Items are balanced in positive and negative keying with negatively keyed items reverse scored. For what we refer to here as traditional or conventional dichotomous scoring, polytomous item scores of 6 and 7 are rescored as 1 and scores of 1 to 5 as 0. With this procedure, high scores reflect clearly exaggerated endorsement of socially desirable behaviors [1–3]. For the alternative dichotomous scoring method proposed by Vispoel and Kim [22], additionally see [23], polytomous item scores of 3 to 7 are rescored as 1 and scores of 1 and 2 as 0. Here, low scores reflect clearly exaggerated denial of socially desirable behaviors but maintain the directionality of polytomous and traditional dichotomous scoring.

We assigned 448 college students (79.3% female, 85.3% Caucasian, mean age = 22.1 and medium age = 19.0) at random to two research conditions: (1) fake good (*n* = 224) and (2) fake bad (*n* = 224). As compensation for participating, students received points counting towards their course grades. The study was sanctioned beforehand by the governing Institutional Review Board, and all participants provided their informed consent before completing the measures. In each condition, respondents completed web-based versions of the BIDR. In the fake-good condition, respondents were initially asked to provide honest answers to all questionnaire items. Upon completion of the questionnaire, they were asked to complete them again but to answer items to convey the best possible impression. The protocol was similar in the fake-bad condition except that respondents were told to answer questionnaire items to give the worst possible impression after having answered them

honestly first. Respondents were asked to answer honestly first to provide a clearer baseline for generating faked responses when they answered the items a second time and thereby enhance their chances of successfully faking. Specific directions for honest, fake-good, and fake-bad responding appear below.

**Honest Responding Directions:** When responding to these survey items, we would like you to be as honest as possible. That is, present yourself as you really are.

**Fake-Good Directions:** When responding to the items, we would like you to try to give the best possible impression of yourself. That is, please respond so as to present yourself in the best possible light. For example, imagine that you are applying for a job for which you strongly desire to be hired. Answer in such a way as to make yourself seem like the best possible applicant for that job.

**Fake-Bad Directions:** When responding to the items, we would like you to try to give the worst possible impression of yourself. That is, please respond so as to present yourself in the worst possible light. For example, imagine that you are being required to apply for a job for which you absolutely do not want to be hired. Answer in such a way as to make yourself seem like the worst possible applicant for that job.

At the end of each questionnaire, we asked participants to paraphrase the directions they received and describe the strategy they used to respond to items. The sample of 448 respondents described here represents individuals who provided evidence that they understood the directions. To maintain the independence of honest and faked responses and take full advantage of all collected data in the classification analyses (i.e., include 448 cases in each condition), honest scores from the fake-bad condition were combined with fake-good scores from the fake good condition, and honest scores for the fake-good conditions were combined with fake-bad scores from the fake-bad condition. This allowed for stricter tests of classification accuracy because it would rarely be the case in practice that half of the respondents would willfully fake responses. Supportive evidence of this comes from Schneider [40] who demonstrated that overall classification accuracy improved with reductions in percentages of faked responses using a variety of measures and detection methods.

*4.2. Analyses*

Analyses reported here include reliability estimates (alpha, omega), descriptive statistics (means, standard deviations, skewness indices, standardized mean differences between honest and faking conditions, scale intercorrelations), model fit indicators, and classification accuracy indices. For each scoring method within each research condition (honest, fake good, fake bad), three confirmatory factor models were tested: (1) a single factor model encompassing both IM and SDE item scores, (2) a correlated two-factor model representing IM and SDE item scores separately, and (3) a correlated two-factor model representing IM and SDE item scores with an additional orthogonal method factor for negatively phrased items. Due to the binary or ordinal nature of BIDR scores, we used diagonally weighted least squares estimation (WLSMV in R) in these analyses. Consistent with conventional guidelines [41–43], we considered model fits as acceptable when comparative fit indexes (CFIs) and Tucker-Lewis indexes (TLIs) equaled 0.90 or higher and root mean square errors of approximation (RMSEAs) equaled 0.08 or lower, and as excellent when CFIs and TLIs equaled 0.95 or higher and RMSEAs equaled 0.06 or lower. Within the classification analyses, we determined cut scores for each scoring method that maximized overall classification accuracy, along with corresponding percentages of correct classifications, false positive errors (classifying honest as faked responses), and false negative errors (classifying faked as honest responses).

## 5. Results

*5.1. Reliability, Distributional Indices, and Scale Correlations under Honest and Fake Conditions*

**Reliability coefficients.** Reliability estimates (alpha & omega), means, standard deviations, skewness indices, standardized mean differences (*d* values), and dependent sample

*t* statistics for BIDR scores under honest and fake conditions appear in Table 1. Consistent with previous research within honest responding conditions [22–29], reliability estimates for polytomous scoring of the IM and SDE scales are uniformly higher than those for dichotomous exaggerated endorsement scoring, but this is not always the case for dichotomous exaggerated denial scoring. When using polytomous or dichotomous scoring targeted in the proper direction (exaggerated endorsement for faking good and exaggerated denial for faking bad), reliability coefficients for faking in either direction always match or exceed those for responding honestly.

**Table 1.** Descriptive Statistics and Reliability Estimates within Honest and Faking Conditions.

| Condition/Scale/Scoring Method | Type of Responding/Index | | | | | | | | | | | |
| | Honest Responding Index | | | | | Faked Responding Index | | | | | Effect Size and *t*-Value | |
| | α | ω | M | SD | Skew | α | ω | M | SD | Skew | d | t |
| **Fake-Good Condition (*n* = 224)** | | | | | | | | | | | | |
| Impression Management | | | | | | | | | | | | |
| Polytomous | 0.77 | 0.78 | 80.42 | 15.92 | 0.06 | 0.88 | 0.89 | 119.55 | 19.56 | −1.27 | 1.69 | 25.30 |
| Traditional Dichotomous | 0.73 | 0.74 | 6.34 | 3.44 | 0.60 | 0.91 | 0.92 | 15.36 | 5.07 | −1.53 | 1.67 | 24.95 |
| Alternative Dichotomous | 0.73 | 0.74 | 14.15 | 3.36 | −0.67 | 0.71 | 0.72 | 18.24 | 2.15 | −2.05 | 1.08 | 16.20 |
| Self-Deceptive Enhancement | | | | | | | | | | | | |
| Polytomous | 0.72 | 0.73 | 81.99 | 13.22 | −0.05 | 0.86 | 0.87 | 112.92 | 17.39 | −0.81 | 1.53 | 22.83 |
| Traditional Dichotomous | 0.67 | 0.68 | 5.16 | 3.13 | 0.67 | 0.90 | 0.91 | 13.16 | 5.28 | −1.00 | 1.48 | 22.18 |
| Alternative Dichotomous | 0.73 | 0.74 | 15.69 | 3.16 | −0.68 | 0.65 | 0.66 | 18.20 | 1.97 | −2.30 | 0.77 | 11.49 |
| **Fake-Bad Condition (*n* = 224)** | | | | | | | | | | | | |
| Impression Management | | | | | | | | | | | | |
| Polytomous | 0.75 | 0.76 | 80.59 | 16.19 | −0.17 | 0.85 | 0.86 | 30.25 | 14.59 | 2.06 | −2.26 | −33.09 |
| Traditional Dichotomous | 0.70 | 0.71 | 7.16 | 3.33 | 0.38 | 0.59 | 0.62 | 1.09 | 1.52 | 1.91 | −1.62 | −24.27 |
| Alternative Dichotomous | 0.71 | 0.72 | 13.52 | 3.23 | −0.66 | 0.90 | 0.90 | 2.16 | 3.55 | 2.64 | −2.48 | −37.14 |
| Self-Deceptive Enhancement | | | | | | | | | | | | |
| Polytomous | 0.69 | 0.71 | 81.75 | 13.07 | −0.04 | 0.69 | 0.71 | 56.52 | 17.51 | 0.18 | −1.10 | −16.49 |
| Traditional Dichotomous | 0.67 | 0.67 | 5.29 | 3.14 | 0.38 | 0.67 | 0.69 | 5.12 | 2.91 | 0.54 | −0.04 | −0.56 |
| Alternative Dichotomous | 0.70 | 0.71 | 15.50 | 3.08 | −0.82 | 0.78 | 0.79 | 7.00 | 3.84 | 0.64 | −1.75 | −26.20 |

*Note.* α = alpha coefficient, ω = omega coefficient, *M* = mean, *SD* = standard deviation, *Skew* = skewness index, *d* = standardized mean difference = $(M_{faking} - M_{honest})/((SD_{faking} + SD_{honest})/2)$, and *t* = dependent sample *t* statistic. All *t* ratios, except traditional dichotomous scoring for the Self-Deceptive Enhancement scale within the Fake-Bad condition, are statistically significant well beyond the 0.00001 level.

**Differences in means.** Means and corresponding *d* values in Table 1 reveal noticeable differences in the logically anticipated direction when using polytomous and properly targeted dichotomous scoring to detect faking, with *d* values ranging in absolute value from 1.10 to 2.48. These values would all exceed large effects (>0.80) according to the guidelines suggested by Cohen [44]. Standardized mean differences between honest and faked responses are greater for IM than for SDE and greater for detecting faking bad than for detecting faking good. IM polytomous scoring (*d* = 1.69) and dichotomous exaggerated endorsement scoring (*d* = 1.67) best separate means for detecting faking good and IM dichotomous exaggerated denial (*d* = −2.48) and polytomous scoring (*d* = −2.26) do so for detecting faking bad.

**Differences in standard deviations.** A consistent pattern of relationships between standard deviations is evident between honest and fake-good responses for polytomous and dichotomous exaggerated endorsement scoring with standard deviations for faking good always exceeding those for responding honestly for both IM and SDE (see Table 1). However, the opposite is true for dichotomous exaggerated denial scoring, which yields lower standard deviations when faking good than when responding honestly. Overall, the pattern of relationships for standard deviations between honest and fake-bad responses is less consistent and varies across scales. For IM, standard deviations are lower for polytomous and dichotomous exaggerated endorsement when faking bad than when responding honestly but the opposite is true for dichotomous exaggerated denial scoring.

For SDE, standard deviations for polytomous and dichotomous exaggerated denial scoring are higher when faking bad than when responding honestly, but the reverse holds for dichotomous exaggerated endorsement scoring.

**Differences in skewness.** In keeping with the differences in reliability, central tendency, and variability already noted, faking behavior noticeably affects the skewness of score distributions. When responding honestly, skewness indices vary from $-0.17$ to $0.06$ for polytomous scoring, from $0.38$ to $0.68$ for dichotomous exaggerated endorsement scoring, and from $-0.82$ to $-0.66$ for dichotomous exaggerated denial scoring. However, when faking good, skewness indices for all scoring methods are negative in sign and increase markedly in absolute value, varying from $-2.05$ to $-1.27$ for IM and from $-2.30$ to $-0.81$ for SDE. Negative skewness indicates that outlying scores are at the lower end of distribution, whereas most scores are at the higher end. When faking bad, the general trend is in the opposing direction, with skewness indices for all scoring methods being positive in sign and typically higher in absolute value than when responding honestly. Specifically, skewness values for faking bad range from $1.91$ to $2.34$ for IM and $0.18$ to $0.64$ for SDE. Positive skewness indicates that outlying scores are at the higher end of the distribution and most scores are at the lower end.

**Scale intercorrelations.** Table 2 includes intercorrelations among BIDR subscale scores for the three scoring methods within honest, fake-good, and fake bad conditions. For honest responding within the same subscale, correlation coefficients between the polytomous and the two dichotomous scoring methods equal $0.86$ and $0.84$ for IM and $0.75$ and $0.75$ for SDE. These values are noticeably higher that those between the two dichotomous scoring methods that equal $0.49$ for IM and $0.17$ for SDE. Correlation coefficients for the same scoring method between IM and SDE, respectively, equal $0.38$, $0.33$, and $0.42$ for polytomous, dichotomous exaggerated endorsement and dichotomous exaggerated denial scores. Overall, these relative differences are highly congruent with those reported by Vispoel and Kim [22] under honest responding conditions.

**Table 2.** Correlations Between IM and SDE Scores in Honest, Fake-Good, and Fake-Bad Conditions.

| | IM-Poly | IM-DEE | IM-DED | SDE-Poly | SDE-DEE |
|---|---|---|---|---|---|
| **Honest (*n* = 448)** | | | | | |
| IM-Poly | | | | | |
| IM-DEE | 0.86 | | | | |
| IM-DED | 0.84 | 0.49 | | | |
| SDE-Poly | 0.38 | 0.34 | 0.30 | | |
| SDE-DEE | 0.21 | 0.33 | 0.01 | 0.75 | |
| SDE-DED | 0.34 | 0.16 | 0.42 | 0.75 | 0.17 |
| **Fake Good (*n* = 224)** | | | | | |
| IM-Poly | | | | | |
| IM-DEE | 0.96 | | | | |
| IM-DED | 0.70 | 0.51 | | | |
| SDE-Poly | 0.70 | 0.71 | 0.34 | | |
| SDE-DEE | 0.71 | 0.78 | 0.24 | 0.94 | |
| SDE-DED | 0.20 | 0.10 | 0.42 | 0.51 | 0.23 |
| **Fake Bad (*n* = 224)** | | | | | |
| IM-Poly | | | | | |
| IM-DEE | 0.84 | | | | |
| IM-DED | 0.96 | 0.71 | | | |
| SDE-Poly | 0.32 | 0.25 | 0.32 | | |
| SDE-DEE | $-0.09$ | 0.03 | $-0.13$ | 0.81 | |
| SDE-DED | 0.53 | 0.35 | 0.57 | 0.90 | 0.50 |

*Note.* IM = Impression Management scale, SDE = Self-Deceptive Enhancement scale, Poly = polytomous scoring, DEE = dichotomous exaggerated endorsement scoring, DED = dichotomous exaggerated denial scoring.

For faking good, correlations of scores across scales and scoring methods are generally higher than for honest responding ($M = 0.54$ vs. $0.42$), and this is especially true for

correlations between IM and SDE for scoring methods oriented in the proper direction (i.e., 0.70 vs. 0.38 for polytomous and 0.78 vs. 0.33 for dichotomous exaggerated endorsement). For faking bad, the mean correlation coefficient across scales and scoring methods is the same (0.42) as for honest responding, and correlations between IM and SDE for scoring methods oriented in the proper direction are inconsistent and vary less with those for honest responding (i.e., 0.32 vs. 0.38 for polytomous and 0.50 vs. 0.17 for dichotomous exaggerated denial). For both IM and SDE subscales within all conditions, polytomous scores are more highly correlated with dichotomous scores than are dichotomous scores with each other. Overall, results in this section serve to emphasize that correlational patterns differ across honest, fake-good, and fake-bad responding conditions, that polytomous scoring captures most of the information provided by both dichotomous scoring methods across conditions, and that results from the two dichotomous scoring methods share the least in common.

### 5.2. Confirmatory Factor Analysis Results under Honest and Faking Conditions

In Table 3, we report confirmatory factor analysis results for the three models mentioned earlier: (1) single factor, (2) two correlated factors representing IM and SDE scores, and (3) two correlated factors representing IM and SDE scores plus an additional orthogonal method factor for negatively worded items. These models were tested for each of the three scoring methods (polytomous, dichotomous exaggerated endorsement, dichotomous exaggerated denial) under honest, fake-good, and fake-bad conditions.

**Table 3.** Model Fit Statistics for BIDR Scores.

| | Condition and Fit Index | | | | | | | | |
|---|---|---|---|---|---|---|---|---|---|
| **Scoring Method and Model** | **Honest (*n* = 448)** | | | **Fake Good (*n* = 224)** | | | **Fake Bad (*n* = 224)** | | |
| | **CFI** | **TLI** | **RMSEA** | **CFI** | **TLI** | **RMSEA** | **CFI** | **TLI** | **RMSEA** |
| **Polytomous** | | | | | | | | | |
| 1. Single factor | 0.69 | 0.68 | 0.06 | 0.96 | 0.96 | 0.06 | 0.95 | 0.95 | 0.05 |
| 2. Two correlated factors | 0.75 | 0.73 | 0.06 | 0.97 | 0.97 | 0.05 | 0.95 | 0.95 | 0.05 |
| 3. Two correlated factors plus one orthogonal method factor for negatively worded items | 0.79 | 0.77 | 0.05 | 0.97 | 0.97 | 0.05 | 0.95 | 0.95 | 0.05 |
| **Dichotomous exaggerated endorsement** | | | | | | | | | |
| 1. Single factor | 0.68 | 0.66 | 0.04 | 0.97 | 0.97 | 0.04 | 0.6 | 0.58 | 0.04 |
| 2. Two correlated factors | 0.77 | 0.75 | 0.04 | 0.98 | 0.98 | 0.03 | 0.64 | 0.62 | 0.04 |
| 3. Two correlated factors plus one orthogonal method factor for negatively worded items | 0.86 | 0.85 | 0.03 | 0.99 | 0.98 | 0.03 | 0.72 | 0.70 | 0.03 |
| **Dichotomous exaggerated denial** | | | | | | | | | |
| 1. Single factor | 0.80 | 0.78 | 0.04 | 0.85 | 0.84 | 0.03 | 0.94 | 0.94 | 0.04 |
| 2. Two correlated factors | 0.86 | 0.85 | 0.03 | 0.85 | 0.84 | 0.03 | 0.95 | 0.95 | 0.03 |
| 3. Two correlated factors plus one orthogonal method factor for negatively worded items | 0.92 | 0.91 | 0.02 | 0.91 | 0.9 | 0.02 | 0.95 | 0.95 | 0.03 |

*Note.* CFI: Comparative Fit Index, TLI: Tucker-Lewis Index, RMSEA: Root Mean Square Error of Approximation.

Within the honest responding condition for all scoring methods, the two-correlated factor models fit better than the single-factor model, and the fit for the two-factor model further improved when adding an orthogonal method factor for negatively worded items. However, that best fitting model only yielded an adequate fit to the data for dichotomous exaggerated denial scoring (CFI = 0.92, TLI = 0.91, RMSEA = 0.02). Although inadequate model fits for polytomous and dichotomous exaggerated endorsement scores do not invalidate the use of those scores for detecting faking, they do raise questions about the

dimensionality of BIDR scores under honest responding conditions (see Gignac [45] and Paulhus & Reid [46], who discuss additional possible constructs measured by BIDR items).

Within appropriately targeted faking conditions, model fit improved dramatically for all scoring methods (CFI = 0.94–0.99, TLI = 0.94–0.98, & RMSEA = 0.03–0.06), with the single-factor model fitting the data nearly as well as both two-factor models. Adding the method factor to the two-factor models yielded negligible or no further improvements in fit. Overall, these results suggest that constructs measured by the BIDR change under each faking condition, likely representing unidimensional response sets that permeate responses across the IM and SDE scales.

### 5.3. Classification Accuracy

**Faking good.** Table 4 includes cut scores that maximize overall classification accuracy, along with percentages of overall correct classifications, false positive errors (classifying honest as faked), and false negative errors (classifying faked as honest) for both types of faking. Classification accuracy in separating honest from fake-good responses using the IM scale is very similar for polytomous (87.95%) and traditional dichotomous scoring (87.72%), and noticeably worse for alternative dichotomous scoring that emphasizes denial rather than endorsement of socially desirable behaviors (79.24%). The percentages of false positive errors for polytomous and dichotomous endorsement scoring are very low (1.34% and 1.12%) in comparison with false negative errors (10.72% and 11.16%). However, the higher percentage of false negative errors is largely attributable to many respondents being unable to fake effectively even when providing evidence that they understood the directions. For example, the IM polytomous mean score for respondents with false positive errors was 87.98 compared to 128.16 for respondents correctly classified as faking good.

**Table 4.** Cut Scores and Classification Accuracy for BIDR Scales and Scoring Methods.

| Condition/Scale/Scoring Method | Index | | | |
|---|---|---|---|---|
| | Cut Score | % Correct Classification | % False Positive | % False Negative |
| **Fake-Good Condition (*n* = 224)** | | | | |
| Impression Management | | | | |
| Polytomous | 109 | 87.95 | 1.34 | 10.71 |
| Traditional Dichotomous | 14 | 87.72 | 1.12 | 11.16 |
| Alternative Dichotomous | 18 | 79.24 | 8.71 | 12.05 |
| Self-Deceptive Enhancement | | | | |
| Polytomous | 99 | 85.27 | 4.24 | 10.49 |
| Traditional Dichotomous | 11 | 85.27 | 2.68 | 12.05 |
| Alternative Dichotomous | 17 | 70.76 | 22.99 | 6.25 |
| **Fake-Bad Condition (*n* = 224)** | | | | |
| Impression Management | | | | |
| Polytomous | 48 | 93.30 | 1.12 | 5.58 |
| Traditional Dichotomous | 2 | 89.51 | 3.79 | 6.70 |
| Alternative Dichotomous | 5 | 93.08 | 0.89 | 6.03 |
| Self-Deceptive Enhancement | | | | |
| Polytomous | 68 | 80.36 | 7.81 | 11.83 |
| Traditional Dichotomous | 4 | 53.57 | 22.10 | 24.33 |
| Alternative Dichotomous | 11 | 87.50 | 6.03 | 6.47 |

In comparison to the IM scale, overall classification accuracy is lower for the SDE scale, but polytomous (85.27%) and dichotomous exaggerated endorsement (85.27%) scoring again provide greater classification accuracy than does dichotomous exaggerated denial (70.67%) scoring. The percentages of false negative errors for polytomous and dichotomous endorsement scoring (10.49% and 12.05%) again exceed percentages of false positive er-

rors (4.24% and 2.68%), but this again is largely attributable to many respondents being unsuccessful in faking good. Here, for example, the mean polytomous score equals 120.37 for respondents correctly classified as faking good versus 84.85 for respondents with false negative errors.

**Faking Bad.** Results for the IM scale within the fake-bad condition in Table 2 reveal that polytomous (93.30%) and alternative dichotomous (93.03) scoring emphasizing exaggerated denial of socially desirable behaviors provide the most accurate classifications for detecting faking bad, and traditional dichotomous (89.51%) scoring emphasizing exaggerated endorsement of such behaviors provides the least accurate classifications. Nevertheless, overall classification accuracy for all three scoring methods exceeds that for the fake-good condition. Percentages of false negative errors again are greater than percentages of false positive errors but are lower overall in comparison to faking good.

As was the case with faking good, the SDE scale is less effective in separating honest from faked responses than is the IM scale. For SDE, dichotomous exaggerated denial (87.50%) scoring provides greater classification accuracy than do either polytomous (80.39%) or dichotomous exaggerated endorsement (53.57%) scoring. The noticeably lower accuracy for dichotomous exaggerated endorsement scoring of the SDE scale indicates that it is ill suited for detecting faking bad. For the most effective scoring methods in detecting faking bad, percentages of false negative errors again are higher than percentages of false positive errors, but lower in comparison to faking good. These reduced percentages of false negative errors for both scales provide some indication that respondents tend to be better at faking bad than faking good.

## 6. Summary and Discussion

Valid interpretation of results from self-report measures is seriously compromised whenever respondents answer items inaccurately by overly endorsing or denying socially desirable behaviors. Our goals in the study reported here were to compare BIDR IM and SDE subscale scores in psychometric properties under honest and faking conditions; determine the effectiveness of polytomous, conventional dichotomous, and alternative dichotomous scoring of BIDR scales when using cut scores to detect faked responses; and estimate the best cut scores to use for those purposes.

The results revealed that respondents in general could successfully fake answers to look either good or bad when asked to do so, and that such faking behavior, in most instances, had noticeable effects on subscale score means, standard deviations, shapes of distributions, score intercorrelations, and model fit. Effects of faking were more pronounced when scoring was oriented in the proper direction (i.e., exaggerated endorsement for detecting faking good and exaggerated denial for detecting faking bad), typically leading to more extreme mean scores, greater score variability, exaggerated skewness, higher inter-subscale correlations, and stronger overall model fits. Accordingly, response sets due to faking were also better detected when scoring was oriented in the proper direction.

Consistent with Paulhus et al. [30], the IM scale was more effective than the SDE scale in separating honest from faked responses within both fake-good and fake-bad conditions. Although no method was completely foolproof, traditional dichotomous and polytomous scoring performed well and with comparable accuracy in detecting faking good, whereas alternative dichotomous and polytomous scoring worked best in detecting faking bad. The only instance in which dichotomous scoring provided better results than polytomous scoring was when using the SDE scale to detect faking bad. However, that scale was less effective in detecting faking bad that was the IM scale for which polytomous and dichotomous exaggerated denial scoring provided comparable results.

Taken collectively, these findings indicate that either polytomous or combinations of dichotomous scoring can be effective in detecting both types of faking. Among the scores considered here, those from the BIDR's IM scale provided the best means to detect either type of faking with polytomous scoring yielding classification accuracy comparable to any combination of dichotomous scores. Consequently, inclusion of that scale along

with targeted questionnaires of interest may provide the most practical way to screen for possible instances of faking either good or bad when using these measures. In the absence for locally available data relevant to fake detection, researchers and practitioners can use the cut scores derived here as possible guides to screen for possible faking when using the BIDR.

As a final note, we emphasize that the present results are limited to college students who completed measures in low-stakes situations for which they were compensated by receiving extra credit within the classes from which they were recruited. Informative future studies in fake detection might focus on different types of respondents in varying settings, directions targeted to more specific situations (e.g., applications for specific jobs; see, e.g., [30,47], use of item-response-theory-based in addition to classical-test-theory-based scores [22,25,26], comparisons with other detection methods [40], and determination of how classification accuracy is affected when using briefer versions of the scales considered here (see, e.g., [25,47–49]).

**Author Contributions:** Conceptualization, W.P.V.; methodology, W.P.V. and M.K.; software, M.K.; validation, W.P.V., M.K. and W.S.S.; formal analysis, W.P.V., M.K. and W.S.S.; investigation, W.P.V.; resources, W.P.V. and M.K.; data curation, W.P.V.; writing—original draft preparation, W.P.V.; writing—review and editing, W.P.V.; visualization, W.P.V., M.K. and W.S.S.; supervision, W.P.V.; project administration, W.P.V. All authors have read and agreed to the published version of the manuscript.

**Funding:** This research received no external funding.

**Institutional Review Board Statement:** This study was conducted in accordance with the Declaration of Helsinki and approved by the Institutional Review Board of the University of Iowa (ID# 20080987380).

**Informed Consent Statement:** Informed concern was obtained from all subjects involved in the study.

**Data Availability Statement:** Inquiries concerning the data should be directed to the lead author.

**Acknowledgments:** The authors thank Delroy Paulhus for granting us permission to administer the BIDR in computerized form for use in this study, Carrie A. Morris for her assistance with data collection, and Tingting Chen for her help in proofreading our submissions.

**Conflicts of Interest:** The authors declare that they have no conflict of interest.

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
