# Peer review of "Detecting Faking on Self-Report Measures Using the Balanced Inventory of Desirable Responding"

_psych, doi:10.3390/psych5040074_

Round 1
Reviewer 1 Report
The primary focus of the manuscript is the identification of deceptive responses on self-report measures through the utilization of the Balanced Inventory of Desirable Responding (BIDR). The BIDR is a crucial tool for assessing socially desirable responses in self-reports. The findings and report may be of interest to survey researchers and professionals, such as human resources practitioners.
Overall, this paper is a well-written and informative contribution to measuring socially desirable responses and scoring of the BIDR. The authors have provided a clear and concise overview of the BIDR and related research results. The aims of the study are well-formulated, and the study design and data analysis appear to be rigorous and appropriate.
I just have a few small comments:
1. It would be worthwhile to consider briefly presenting the results of the confirmatory factor analysis (CFA) of BIDR. It can be assumed that the 2-factor model works well and would highlight the generalizability of the results and the acceptability of the psychometric properties of the measure.
2. In the discussion section, it would be useful to formulate some limitations, e.g., the participation of university students, the fact that the questionnaires were completed in a timely and sequential manner, and the fact that everyone received credit for participation in the research in a uniform way.
3. In the discussion section, it would be useful to relate the results and their interpretation back to the existing literature.
4. It would also be useful to illustrate the correlations between the scales.
In my opinion, the above-mentioned additional notes and analysis would greatly increase the value of this study.
Congratulations to the authors for an interesting and valuable study.
Reviewer 2 Report
The present manuscript examines the role of the response format/scoring methods of the frequently used Balanced Inventory of Desirable Responding in a sample of 224 participants and different conditions concerning instructions regarding faking.
Overall, the manuscript is interesting and fits into the scope of psych. I found the manuscript well written and methodologically sound. Upon my reading I only found three minor points that I recommend to address before publication. Congratulations on the interesting study.
(1) Please add a data analysis section at the end of the methods section. This helps readers to understand how the research questions are addressed methodologically.
(2) Please consider adding McDonald's omega estimates in addition to the alpha coefficients to allow for internal consistency estimations that are less restricted by the assumptions often not met for alpha.
(3) The comparison of the mean and SD values across conditions and versions assumes that the measurement model is the same. Please consider testing the measurement models with CFAs. Since the data include dichotomous response formats, please use the WLSMV estimator (see Brauer et al., 2023, for a discussion).
Brauer, K., Ranger, J., & Ziegler, M. (2023). Confirmatory factor analyses in Psychological Test Adaptation and Development: A non-technical discussion of the WLSMV estimator. Psychological Test Adaptation and Development, 4, 4-12.
